# Clinical and Immunologic Characteristics of Colorectal Cancer Tumors Expressing LY6G6D

**DOI:** 10.3390/ijms25105345

**Published:** 2024-05-14

**Authors:** Adrián Sanvicente García, Manuel Pedregal, Lucía Paniagua-Herranz, Cristina Díaz-Tejeiro, Cristina Nieto-Jiménez, Pedro Pérez Segura, Gyöngyi Munkácsy, Balázs Győrffy, Emiliano Calvo, Víctor Moreno, Alberto Ocaña

**Affiliations:** 1Experimental Therapeutics in Cancer Unit, Hospital Clínico San Carlos (HCSC), Instituto de Investigación Sanitaria San Carlos (IdISSC), 28040 Madrid, Spain; adriansanvicenteg@gmail.com (A.S.G.); lucia.paniagua@salud.madrid.org (L.P.-H.); cristina.diaztejeiro@salud.madrid.org (C.D.-T.); cnietoj@salud.madrid.org (C.N.-J.); 2Facultad de Ciencias Químicas, Universidad Complutense de Madrid, 28040 Madrid, Spain; 3START Madrid-Fundación Jiménez Díaz (FJD) Early Phase Program, Fundación Jiménez Díaz Hospital, 28040 Madrid, Spain; manuel.pedregal@startmadrid.com (M.P.); emiliano.calvo@startmadrid.com (E.C.); victor.moreno@startmadrid.com (V.M.); 4Medical Oncology Department, Hospital Clínico San Carlos (HCSC), Instituto de Investigación Sanitaria San Carlos (IdISSC), 28040 Madrid, Spain; pedro.perez@salud.madrid.org; 5Department of Bioinformatics, Semmelweis University, H-1094 Budapest, Hungary; munkacsy@gyer1.sote.hu (G.M.); gyorffy.balazs@yahoo.com (B.G.); 6Department of Pediatrics, Semmelweis University, H-1094 Budapest, Hungary; 7Research Centre for Natural Sciences, Institute of Enzymology, H-1117 Budapest, Hungary; 8Department of Biophysics, Medical School, University of Pecs, H-7624 Pecs, Hungary; 9START Madrid-HM Centro Integral Oncológico Clara Campal (CIOCC), Early Phase Program, HM Sanchinarro University Hospital, 28050 Madrid, Spain; 10Centro de Investigación Biomédica en Red Cáncer (CIBERONC), 28029 Madrid, Spain

**Keywords:** LY6G6D, CRC, TCEs, immune association

## Abstract

The identification of targets that are expressed on the cell membrane is a main goal in cancer research. The Lymphocyte Antigen 6 Family Member G6D (*LY6G6D*) gene codes for a protein that is mainly present on the surface of colorectal cancer (CRC) cells. Therapeutic strategies against this protein like the development of T cell engagers (TCE) are currently in the early clinical stage. In the present work, we interrogated public genomic datasets including TCGA to evaluate the genomic and immunologic cell profile present in tumors with high expression of *LY6G6D*. We used data from TCGA, among others, and the Tumor Immune Estimation Resource (TIMER2.0) platform for immune cell estimations and Spearman correlation tests. LY6G6D expression was exclusively present in CRC, particularly in the microsatellite stable (MSS) subtype, and was associated with left-side tumors and the canonical genomic subgroup. Tumors with mutations of *APC* and *p53* expressed elevated levels of LY6G6D. This protein was expressed in tumors with an inert immune microenvironment with an absence of immune cells and co-inhibitory molecules. In conclusion, we described clinical, genomic and immune-pathologic characteristics that can be used to optimize the clinical development of agents against this target. Future studies should be performed to confirm these findings and potentially explore the suggested clinical development options.

## 1. Introduction

The identification of targets expressed on the surface of the cell membrane is a main goal in cancer research to develop novel therapeutic strategies [1]. The selection of tumor-associated antigens (TAAs) specifically expressed in tumoral cells can permit selective targeting avoiding on-target off-tumor toxicities [2]. TAAs can be used for the development of different antibody-based modalities like antibody-drug conjugates (ADCs), bi-specific antibodies, or T cell engagers (TCE) [2,3,4].

All these strategies have demonstrated clinical efficacy in solid and hematological malignancies. At this moment, there are eleven ADCs, one bi-specific and two TCEs approved by the US Food and Drug Administration (FDA) [5]. It is envisioned that several other compounds will reach the clinic in the following years. Considering the high grade of toxicity observed with some of these agents, particularly TCE, it would be desirable to find cancer types and indications where TAAs are exclusively expressed in tumors [4]. Compared with hematological malignancies where monoclonal expansions of tumor cells are the standard, in solid tumors the identification of specific TAAs is exceptional due to the high grade of tumor heterogenicity [6].

The implementation of genomic techniques complemented with proteomic studies has permitted the identification of novel targets that could be considered optimal TAAs for antibody development [7]. Our group, based on these approaches, has recently described some novel membrane proteins for the development of ADCs [8,9]. Additional examples have reached early clinical development like those acting on the Lymphocyte Antigen 6 Family Member G6D (*LY6G6D*) gene.

LY6G6D belongs to a cluster of leukocyte antigens located in the major histocompatibility complex (MHC) class III region on chromosome 6 [10]. LY6G6D, like most members of the family, is attached to the cell membrane by a glycosylphosphatidylinositol (GPI) anchor [10,11].

LY6G6D was found to be expressed in colorectal cancer (CRC), a disease whose incidence has dramatically increased over the last years, particularly early-onset colon and rectal cancer where rates are expected to increase by more than 25% and 45%, respectively, by 2030 [12]. Therefore, the identification of novel targets in this disease and the optimization of therapies designed against them is key to better selecting the right patient population [13,14]. A very recent article describes the presence of LY6G6D by immunohistochemistry (IHC) in MSS CRC with a total of 107 primary samples the overall prevalence of LY6G6D expression was 74% (IHC 1þ/2þ/3þ), with moderate to strong LY6G6D expression (IHC 2þ/3þ) in 25% of the cases [14].

In the present work, we aimed to evaluate the genomic profile present in tumors with high expression of LY6G6D at a transcriptomic level. We observed how LY6G6D expression was exclusively present in CRC, particularly in the microsatellite stable (MSS) subtype, and associated with left-side tumors and the canonical subgroup. In addition, tumors with mutations of *APC* and *p53* expressed elevated levels of LY6G6D. Finally, LY6G6D was expressed in tumors with an inert immune microenvironment with an absence of immune cells and co-inhibitory molecules.

## 2. Results

### 2.1. LY6G6D Is Highly and Exclusively Expressed in Colorectal Cancer Tumors

We first aimed to evaluate the transcriptomic presence of LY6G6D in all solid tumors by mapping their expression using publicly available datasets. LY6G6D was highly and exclusively expressed in rectal and colon adenocarcinoma as shown in Figure 1 with an expression of more than 32 transcripts per million (TPM). Of note, the highest expression was observed in rectum adenocarcinoma with more than 100 TPM. Expression in normal tissue was extremely low in all solid tumors with less than 5 TPM (Figure 1A). The fold change of the expression of LY6G6D for each cancer type can be visualized in Figure 1B.

In addition, as age and sex are considered important factors to take into consideration in this disease, especially for early-onset tumors, we decided to test whether there was an association between LY6G6D expression and these factors. As can be seen in Figure 1C, no correlation was observed between gene expression and patient age. Similarly, there were also no significant differences between sexes in relation to *LY6G6D* expression (Figure 1D).

In line with the previous data [14], LY6G6D was highly present in the distal colon when compared with the proximal (Figure 2A), and in the left side and rectum compared with the right side (Figure 2B).

### 2.2. LY6G6D Is More Expressed in the Microsatellite Stable and the Canonical Colon Cancer Subtype

Immune checkpoint inhibitors in colon cancer have only demonstrated activity in microsatellite unstable high (MSI-H) tumors, and few signs of clinical activity have been observed in microsatellite stable (MSS) cancers. Therefore, we explored the association of *LY6G6D* with these tumor subtypes. As can be seen in Figure 2C, LY6G6D was highly present in MSS tumors compared with MSI-H ones. Next, we evaluated the presence of LY6G6D with the different described genomic colon cancer subtypes [15]. The canonical subtype was the one with the highest presence of *LY6G6D* followed by the mesenchymal subgroup (Figure 2D).

### 2.3. Transcriptomic Profiling of Colon Adenocarcinomas (CRC) with High Expression of LY6G6D and Association with Immune Infiltrates

To better characterize the genomic profile of tumors with a high expression of LY6G6D, we evaluated at a transcriptomic level those genes that were upregulated when LY6G6D was highly expressed. Using a cut-off level of FC ≥ 2 and a *p*-value < 0.05, we identified 201 genes that were upregulated in MSI-H and 275 upregulated genes in MSI-L/MSS. Nineteen of these genes were shared between MSI-H and MSI-L/MSS patients (Appendix A).

Using an exclusion criteria with a Spearman correlation cut-off ≥0.4 we found only five genes were shared between subtypes that correlated with LY6G6D.

Then we aimed to evaluate the presence of immune populations when these transcripts were highly present. CD8+, CD4+ T cells and neutrophils correlated negatively with the majority of the selected genes and none of them correlated positively with any immune population (Appendix A). Although these correlations were not strong, these data suggest the presence of an immunosuppressive microenvironment when these genes are upregulated.

### 2.4. Mutational Profile of Tumors with High Expression of LY6G6D

Considering those genes with higher mutational rates in CRC (Figure 3A) we evaluated the correlation of these mutations with LY6G6D expression. A positive and strong association was observed between high LY6G6D and: APC, tp53 and DMD mutations (0.721; 0.536 and 0.428, respectively) (Figure 3B). Contrary, a negative association was identified for KRAS and DNAH5 (−0.195 and −0.299, respectively) (Figure 3B). A weak but positive correlation was observed for HMCN1, FAT4, LRP1B, ERBB2, and LRP2. For the other mutations analyzed no association was observed (Figure 3B).

### 2.5. LY6G6D Is Present in an Inert Immune Microenvironment

Given the role of LY6G6D as a TAA, we studied the type of immune cells that were present in the tumor microenvironment when tumors displayed high expression of LY6G6D. As can be seen in Figure 4A, LY6G6D correlated negatively with some adaptive cell types, including T Cell CD8+ and B cells (Rho = −0.308, *p*-value = 1.88 × 10^−7^ and Rho = −0.170, *p*-value = 4.64 × 10^−3^, respectively) and with selected innate cell types: neutrophils, dendritic cells and Nk cells (Rho = −0.355, *p*-value = 1.4 × 10^−9^; Rho = −0.267, *p*-value = 3.51 × 10^−6^; Rho = −0.250, *p*-value = 2.69 × 10^−5^, respectively). Mast cells, which are important for both innate and adaptative response also correlated negatively (Rho = −0.269, *p*-value = 6.28 × 10^−6^).

For other immune populations such as CD4+ T cells and macrophages, no association was found. In line with this, we explored the association of LY6G6D with T cell activating genes including perforin, granzyme A and B, CD8a, CD8b and PRF1. Similarly, a negative association was observed for CD8A, PRF1 and GZMA (Rho = −0.370, *p*-value = 1.21 × 10^−14^; Rho = −0.353, *p*-value = 2.25 × 10^−13^ and Rho = −0.415, *p*-value = 2.73 × 10^−18^, respectively) (Figure 5A). Similarly, the relationship between APC mutation and the expression of these genes again showed a negative correlation for all but GZMB which did not correlate (Figure 5B).

Finally, we evaluated the presence of LY6G6D with different inhibitory molecules. As displayed in Figure 6A, the expression of LY6G6D correlated negatively with PD1, PD-L1, TIM3, TIGIT and 4-1BB (Rho = −0.326, *p*-value = 1.71 × 10^−11^; Rho = −0.395, *p*-value = 1.32 × 10^−16^; Rho = −0.265, *p*-value = 6.13 × 10^−8^; Rho = −0.331, *p*-value = 8.12 × 10^−12^ and Rho = −0.186, *p*-value = 1.62 × 10^−4^ respectively). Detailed dot plots of all these data are provided in the last section of Figure 4, Figure 5 and Figure 6, respectively.

## 3. Discussion

In our study, we describe the exclusive presence of LY6G6D in CRC particularly in the MSS subtype. These data confirm the published information using transcriptomic and immunohistochemistry (IHC) data [13,14,16] and extend this information to additional biological parameters that could help to define the best target population and combinational strategies for the correct clinical development of agents against this target. In this context, we also observed a clear association with the canonical subtype and an increased expression of LY6G6D in left-sided/rectum tumors, and those harboring APC/P53 mutations.

It is well known that the canonical subtype is more present in left-sided tumors and those with APC/P53 mutations. Both mutations, P53 and APC are implicated, as tumor suppressor genes, in numerous crucial signaling pathways and biological processes associated with chromosomal instability (CIN) and CRC carcinogenesis [17]. As P53 restoration drives tumor regression, some therapeutic strategies have been developed to reactivate its function. For instance, PC14586, a first-in-class p53 reactivator, elicited a response in about a fourth of patients with advanced solid tumors carrying p53 Y220C mutations and demonstrated acceptable tolerability [18]. APC also forms part of the WNT signaling pathway, and its inactivation results in an increase in nuclear CTNNB1 expression and cell proliferation, also playing a gatekeeper role in CIN CRCs [17].

As seen, a clear association was observed for APC and P53, but not KRAS. Indeed, APC mutations correlate with a low presence of immune checkpoint targets and low tumor mutational burden (TMB) [19]. This is also in line with previously described data that suggest that the canonical subtype presents poor infiltration of immune cells [20]. Very weak associations were identified for some of them, for example, mutations at LRP1B/FAT4 were described as potential biomarkers for favorable immunotherapy response independently of the high presence of TMB and MSI-H status [21,22].

Only five transcripts strongly correlated with LY6G6D and were shared between subtypes. These genes did not correlate with any immune population. In line with this, when we explored the tumor microenvironment, we observed a negative correlation between LY6G6D with adaptative immune cells including effector CD8+ T cells. Indeed, markers associated with T cell effector functions were down-regulated. Similarly, innate cells also showed a reduced expression. In this context, the killing of LY6G6D-positive tumoral cells with TCE must take advantage of a limited number of CD8+ T cells. It is unknown if this low level of CD8+ T cells is sufficient for clinical activity, or if additional cells need to be recruited outside the tumoral areas to reach a clinical response.

At this moment there are TCE against LY6G6D in clinical development [13,14]. To our knowledge, no clinical efficacy data have been reported yet. Similarly, no combinatorial strategies are under evaluation with an intent to modify the immunosuppressive microenvironment associated with the presence of LY6G6D.

We acknowledge that this study has limitations. This is a bioinformatic evaluation that uses publicly available genomic data for their analysis. We recognize that studies evaluating the protein content and spatial organization within the tumor microenvironment in the different CRC subtypes should be performed to better characterize the responsive population.

In conclusion, we describe the clinical, genomic, and immune-pathologic characteristics that can be used to optimize the clinical development of agents against this target. The initial findings provided in the present study justify the need for further investigation of LY6G6D in human samples and patient-oriented preclinical models to validate the observed information and identify potential therapeutic combinations.

## 4. Materials and Methods

### 4.1. LY6G6D Expression Studies

Data from TCGA (The Cancer Genome Atlas; https://www.cancer.gov/ccg/research/genome-sequencing/tcga; accessed on 15 December 2022) [23] Pancancer was obtained for this study to achieve information about patient’s gene expression in normal and tumoral tissue in different cancers. This dataset contains whole exome sequencing information from patients’ tumors and their respective healthy tissues. This was assessed using several web tools containing these data as GEPIA2 (http://gepia2.cancer-pku.cn/#index; accessed on 15 December 2022) [24], Gent2 (http://gent2.appex.kr/gent2/; accessed on 15 December 2022) [25] and UALCAN (https://ualcan.path.uab.edu/; accessed on 15 December 2022) [26,27].

We also used Cbioportal to acquire mutational information about several cancers (Colorectal Adenocarcinoma (TCGA, Firehose Legacy)) (https://www.cbioportal.org/; accessed on 16 December 2022) [28].

### 4.2. Transcriptomic Studies

GEO was searched to identify datasets with colon cancer studies with transcriptome-wide gene expression data derived by using Affymetrix gene arrays. Together 2110 samples were identified which stemmed from 16 independent datasets. The average follow-up of these patients was 52.6 months and 47.2% were female. The gene expression data were mas5 normalized and then a second normalization was applied to set the mean expression across all genes to 1000 to reduce batch effects due to different mean expressions in the mas5 normalization.

### 4.3. Identification of Mutated Genes in Tumors with High Expression of LY6G6D

TCGA data were used to investigate gene alterations when LY6G6D was highly expressed in patients with colorectal cancer in MSI-H and MSI-L/MSS independent subsets. Data from these subsets were compared and in order to filter our genes, we set an exclusion criterion based on a Fold Change ≥ 2 and a Spearman correlation ≥ 0.4 between LY6G6D expression and gene mutation. Identified shared genes between subsets were further analyzed to elucidate potential immunogenic roles.

### 4.4. Immune Cell Infiltration and Gene Expression Correlation

To investigate the association between gene expression/mutation and immune infiltration (tumor purity), we used the Tumor Immune Estimation Resource (TIMER2.0) platform [29] (http:/timer.cistrome.org/, accessed on 18 December 2022). TIMER2.0 uses Spearman correlation to associatSe these parameters with immune populations, including T Cell CD4+, T Cell CD8+, B cells, macrophages, dendritic cells and neutrophils. It contains 10,897 samples from diverse cancer types from the TCGA.

### 4.5. Gene Correlations

The Spearman correlation coefficient was used for correlation analysis between gene expression and mutations or immune infiltrates. Statistically significant results (*p*-value < 0.05) were displayed in red or blue, for positive or negative correlations, respectively, using heatmaps. Those with no statistical association were represented in grey. Data from TCGA [23] were used in the analysis.

### 4.6. Graphical Design

Histograms, bar charts, heatmaps, etc., were plotted using GraphPad Prism 10.0.1 software (GraphPad Software, San Diego, CA, USA).

## Figures and Tables

**Figure 1 ijms-25-05345-f001:**
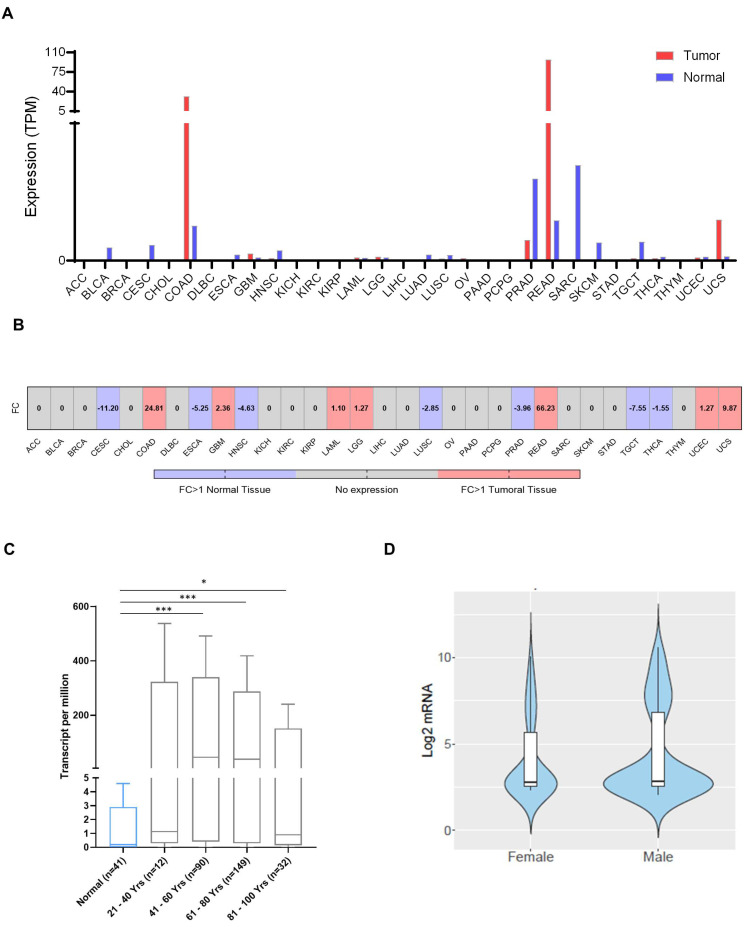
LY6G6D expression profile across several tumor tissues and their paired normal samples. (**A**) LY6G6D expression (Transcript per million—TPM) in different cancer types using GEPIA2 data. (**B**) Fold Change (FC) of the expression of LY6G6D in Normal vs. Tumoral tissue based on GEPIA2 data. (**C**) Correlation between LY6G6D expression (TPM) and patient’s age. UALCAN data. Statistically significant differences between normal and tumoral tissue are marked by an * (* for *p* < 0.05, ***** for *p* < 0.001). (**D**) LY6G6D expression (log2 mRNA) differentiation between patient’s biological gender (F or M). Data obtained from CANCERTOOL. No statistical difference was found.

**Figure 2 ijms-25-05345-f002:**
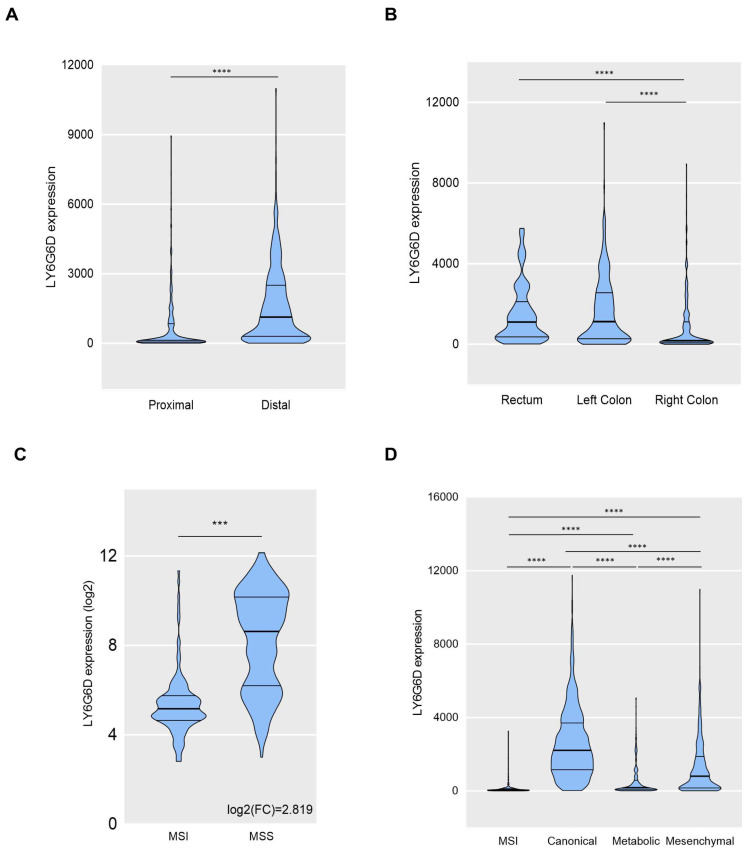
LY6G6D expression analysis in different scenarios. (**A**) LY6G6D expression based on the location of the tumor in the colon (Proximal vs. Distal). (**B**) LY6G6D expression in Rectum, Left colon or Right colon. (**C**) LY6G6D expression depending on microsatellite instability (MSI vs. MSS). (**D**) LY6G6D expression based on molecular subtypes of CRC (Canonical, Metabolic or Mesenchymal). Statistically significant differences are marked as: *** for *p* < 0.01 and **** for *p* < 0.001.

**Figure 3 ijms-25-05345-f003:**
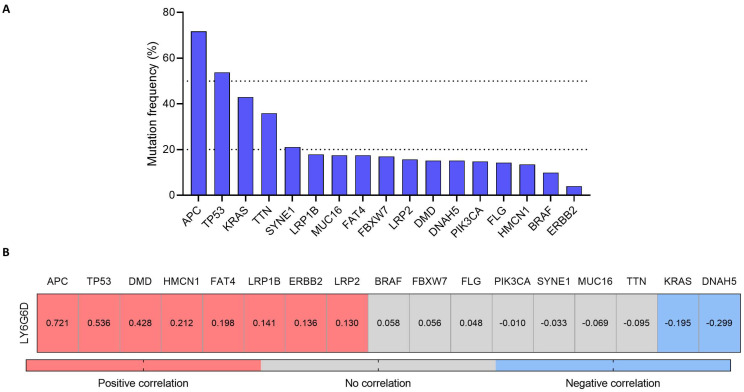
Mutational profile of CRC. (**A**) More frequent mutations in CRC represented as a bar graph. Data obtained from CBioportal (TCGA, Firehose Legacy). (**B**) Correlation between LY6G6D expression and these mutations in CRC. Conditions that are painted in the heatmap, in red for positive correlation and in blue for negative correlation, are statistically significant (*p*-value < 0.05).

**Figure 4 ijms-25-05345-f004:**
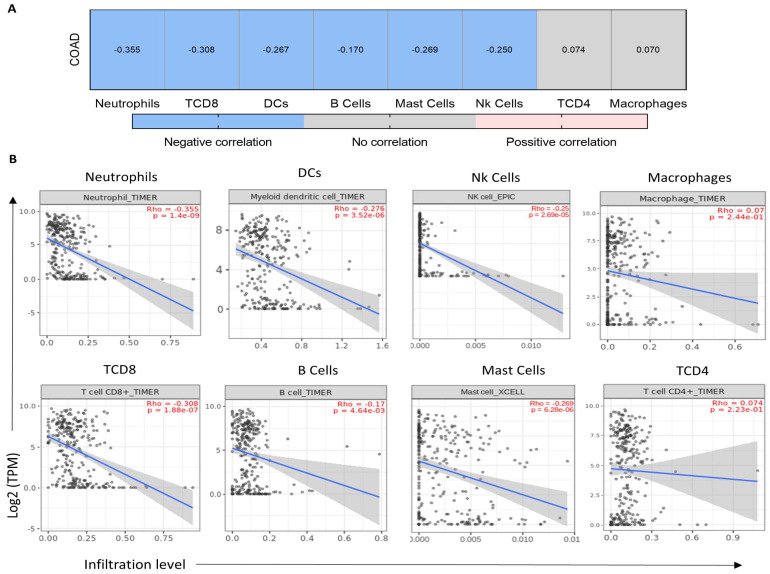
LY6G6D expression level in association with immune populations in CRC. (**A**) Correlation analysis between LY6G6D levels and some relevant immune populations using TIMER2.0. Spearman’s correlation was used with purity adjustment. Conditions that are painted in the heatmap, in red for positive correlation and in blue for negative correlation, are statistically significant (*p*-value < 0.05). (**B**) Dot plot detail of the association displayed in (**A**). LY6G6D expression is represented in the y-axis as log2 (TPM) while infiltration level is in the x-axis.

**Figure 5 ijms-25-05345-f005:**
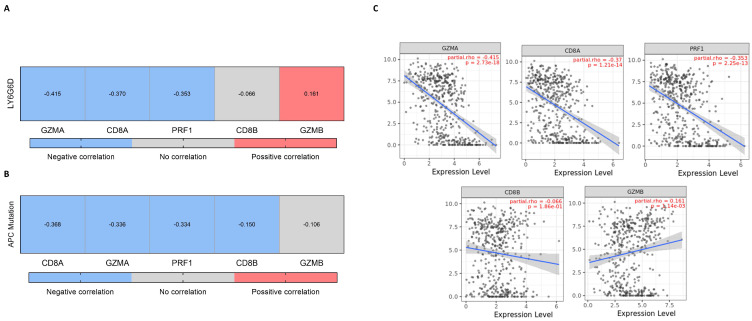
Analysis of the expression of T cell activating genes (TCD8+ signature) in relation to LY6G6D expression or described mutations. (**A**) Heatmap representing the correlation between LY6G6D levels and the expression of the genes in the TCD8+ signature. (**B**) Heatmap showing the correlation between APC mutation and the expression of the cited genes of the TCD8+ signature. Conditions that are painted in the heatmap, in red for positive correlation and in blue for negative correlation, are statistically significant (*p*-value < 0.05) (**C**) Dot plot detail of the association studied in (**A**). LY6G6D expression is represented in the y-axis as log2 (TPM) while the expression level of the other gene is set in the x-axis (log2 (TPM)).

**Figure 6 ijms-25-05345-f006:**
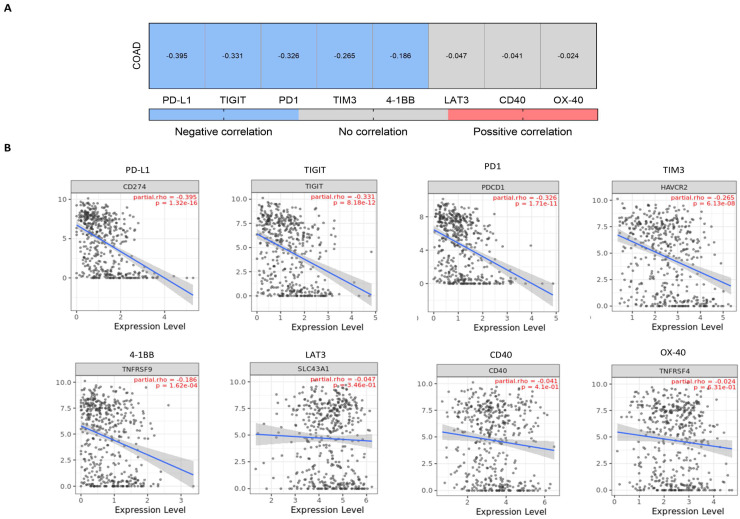
Association of LY6G6D expression levels with co-stimulatory immune checkpoints. (**A**) Heatmap showing the correlation between LY6G6D expression and the expression of selected immunomodulatory genes. Conditions that are painted in the heatmap, in red for positive correlation and in blue for negative correlation, are statistically significant (*p*-value < 0.05). (**B**) Dot plot detail of the association studied in (**A**). LY6G6D expression is represented in the y-axis as log2 (TPM) while the expression level of the co-stimulatory gene is set in the x-axis (log2 (TPM)).

## Data Availability

All sources of data generated or analyzed during this study are included and detailed in the Material and Methods section of this published article. These are openly available in: Gepia2 (http://gepia2.cancer-pku.cn/#index; accessed on 15 December 2022); Gent2 (http://gent2.appex.kr/gent2; accessed on 15 December 2022); CBioportal (TCGA, Firehose Legacy) (https://www.cbioportal.org; accessed on 16 December 2022); TIMER2.0 (http:/timer.cistrome.org; accessed on 18 December 2022); UALCAN (https://ualcan.path.uab.edu/; accessed on 15 December 2022).

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
