# Peer review of "Clinical and Immunologic Characteristics of Colorectal Cancer Tumors Expressing LY6G6D"

_ijms, 2024, doi:10.3390/ijms25105345_

Round 1
Reviewer 1 Report
Comments and Suggestions for Authors
In this manuscript, the authors evaluated the genomic and immunologic cell profile present in tumors with high expression of LY6G6D, using public genomic datasets including TCGA. It focuses on the interesting topic of pursuing the potential of LY6G6D as a tumor-associated antigen as a promising therapeutic target in colorectal cancer (CRC). Some questions and suggestions are listed below.
Specific points)
1. Page 2, lines 67-72; In this paragraph, the previously reported information on the frequency and site of expression of LY6G6D in CRCs should be given in more detail.
2. Page 3, line 99; regarding the description of statistics. Since no significant difference is shown in Figure 1A, the description should be moved to Figure 1C.
3. Page 4, line 104; they should cite the previous paper on the data.
4. Page 5, lines 118-121; no explanation regarding the expression of significant differences (asterisks).
5. Page 5, lines 124-131; It is difficult to understand why MSI-H and MSI-L/MSS were analyzed separately to narrow down the genes associated with high expression of LY6G6D, and then genes common to both were selected.
6. Page 5, line 135; Supplementaryl Fig. 1b, not 1c.
7. Page 5, lines 143-144; the significant correlations were also shown in Figure 3B for HMCN1, FAT4, LRP1B, ERBB2, and LRP2, which contradicts this statement.
8. Page 6, lines 153-157; In Figure 4, there was no significant correlation for CD4+ T cells, so the ALMOST in this statement was an exaggeration.
9. Page 6, line 163; Figure 4A, not Figure 5A.
10. Page 9, lines 218-221; the meaning of this statement is unclear because a significant correlation was shown in Figure 3B for LRP1B/FAT4.
Comments on the Quality of English Language
As for the quality of the English language, only a few expressions were found to be somewhat unclear.
Reviewer 2 Report
Comments and Suggestions for Authors
In this study Sanvincente and colleagues describe the high expression of LY6G6D in some colon cancer patients, particularly in the microsatellite stable subtype and in association with APC and p53 mutation.
The high expression of LY6G6D is not associated with the sex and age of the patients nor with the presence of specific subtypes of cells of the immune system or costimulatory immune checkpoints.
The study concludes indicating LY6G6D as a new molecular marker of CRC, however its presence in an immunologically inert microenvironment could be a limiting factor for its use as immunotherapy target.
The manuscript consists in a bioinformatic evaluation of already published genomic data. Indeed the authors recognize the limitations or their work and suggest the need of further studies to confirm their observations and to identify potential therapeutic approaches.
I would suggest the authors to include also NK and mast cell in the correlation analysis in figure 4, in order to perform a more complete evaluation of the immune tumor microenvironment.
